# Molecular Toxicological Mechanisms of Synthetic Cathinones on C2C12 Myoblasts

**DOI:** 10.3390/ijms20071561

**Published:** 2019-03-28

**Authors:** Xun Zhou, Dino Luethi, Gerda M. Sanvee, Jamal Bouitbir, Matthias E. Liechti, Stephan Krähenbühl

**Affiliations:** 1Division of Clinical Pharmacology & Toxicology, Department of Biomedicine, University Hospital Basel and University of Basel, 4031 Basel, Switzerland; Xun.Zhou@unibas.ch (X.Z.); dino.luethi@unibas.ch (D.L.); Gerda.Sanvee@unibas.ch (G.M.S.); Jamal.Bouitbir@unibas.ch (J.B.); Matthias.Liechti@unibas.ch (M.E.L.); 2Swiss Centre for Applied Human Toxicology, 4031 Basel, Switzerland

**Keywords:** synthetic cathinones, skeletal muscle toxicity, mitochondria, electron transport chain

## Abstract

Synthetic cathinones are popular psychoactive substances that may cause skeletal muscle damage. In addition to indirect sympathomimetic myotoxicity, these substances could be directly myotoxic. Since studies in myocytes are currently lacking, the aim of the present study was to investigate potential toxicological effects by synthetic cathinones on C2C12 myoblasts (mouse skeletal muscle cell line). We exposed C2C12 myoblasts to 3-methylmethcathinone, 4-methylmethcathinone (mephedrone), 3,4-methylenedioxymethcathinone (methylone), 3,4-methylenedioxypyrovalerone (MDPV), alpha-pyrrolidinovalerophenone (α-PVP), and naphthylpyrovalerone (naphyrone) for 1 or 24 h before cell membrane integrity, ATP content, mitochondrial oxygen consumption, and mitochondrial superoxide production was measured. 3,4-Methylenedioxymethamphetamine (MDMA) was included as a reference compound. All investigated synthetic cathinones, as well as MDMA, impaired cell membrane integrity, depleted ATP levels, and increased mitochondrial superoxide concentrations in a concentration-dependent manner in the range of 50–2000 μM. The two pyrovalerone derivatives α-PVP and naphyrone, and MDMA, additionally impaired basal and maximal cellular respiration, suggesting mitochondrial dysfunction. Alpha-PVP inhibited complex I, naphyrone complex II, and MDMA complex I and III, whereas complex IV was not affected. We conclude that, in addition to sympathetic nervous system effects and strenuous muscle exercise, direct effects of some cathinones on skeletal muscle mitochondria may contribute to myotoxicity in susceptible synthetic cathinone drugs users.

## 1. Introduction

Synthetic cathinones are derivatives of cathinone, the naturally occurring alkaloid found in the *Catha edulis* shrub, and represent one of the largest groups of new psychoactive substances (NPS), with more than one hundred different substances detected by the European Union Early Warning System between 2005 and 2017 [1]. As shown in Figure 1, cathinones are structurally closely related to amphetamines, such as 3, 4-methylenedioxymethamphetamine (MDMA, “ecstasy”). The main difference is a β-keto group in the side chain of the cathinones, which is not present in the amphetamines [2]. Even though the frequent use of synthetic cathinones in a recreational setting is a relatively new phenomenon, several of these substances were first synthesized during the twentieth century. For instance, the first synthesis of the frequently abused 4-methylmethcathinone (mephedrone) was published in 1929 [3]. Many additional synthetic cathinones have since been developed, mostly as antidepressant or anorectic agents [4,5,6,7,8,9]. However, due to concerns about abuse liability, only a few of them were marketed.

Synthetic cathinones elicit their pharmacological effect mainly as substrates or inhibitors of monoamine transporters [10,11,12,13,14,15], and their use has been associated with sympathomimetic toxicity, which typically includes tachycardia, hypertension, raised body temperature, diaphoresis, mydriasis and agitation [16,17,18,19,20]. In rare cases, the use of synthetic cathinones may result in significant peripheral organ damage [21,22,23].

Amongst other toxicities, skeletal muscle damage has been linked to the use of cathinone and its synthetic derivatives [23,24,25,26,27], but the underlying mechanisms of toxicity are currently not fully elucidated. Previous studies in human hepatic cells [28,29,30], as well as in dopaminergic human neuroblastoma cells [31,32], showed that synthetic cathinones cause cellular stress and elicit mitochondrial dysfunction, leading to apoptosis and cell death. In order to investigate whether similar toxicological mechanisms also contribute to skeletal muscle toxicity, the in vitro myotoxicity of the synthetic cathinones (Figure 1) 3-methylmethcathinone (3-MMC), 4-methylmethcathinone (4-MMC, mephedrone), 3,4-methylenedioxy-N-methcathinone (methylone), 3,4-methylenedioxypyrovalerone (MDPV), alpha-pyrrolidinovalerophenone (α-PVP), and naphthylpyrovalerone (naphyrone) was studied in C2C12 myoblasts. 3,4-methylenedioxymethamphetamine (MDMA, “ecstasy”) was included in the study as a non-cathinone reference compound with documented myotoxic effects [33,34,35,36,37].

## 2. Results

### 2.1. Cell Viability and ATP Content

The intracellular ATP content was determined for the assessment of mitochondrial function and cell viability and the release of adenylate kinase (AK) as a marker of cell membrane integrity. After treatment for 24 h, all drugs proved to impair cell membrane integrity and to deplete ATP in a concentration-dependent manner in the range of 50–2000 μM (Figure 2). After exposure for 24 h, a significant depletion of ATP at 50 μM was observed for naphyrone, at 200 μM for mephedrone, at 500 μM for methylone, and ≥1 mM for the remaining drugs. The depletion of ATP was accompanied by cell membrane integrity loss for all drugs, reaching significance at slightly higher (methylone and MDPV), or similar concentrations (remaining drugs), at which ATP depletion occurred. The IC50 values for membrane toxicity were higher than for ATP depletion for 3-MMC, mephedrone, α-PVP, MDPV and naphyrone (Table 1).

### 2.2. Effect on Cellular Oxygen Consumption

A decrease in the cellular ATP content in the presence of glucose can result from an impaired function of the mitochondrial respiratory chain or impaired glycolysis [38,39]. In order to determine whether the depletion of ATP mediated by the synthetic cathinones was related to the interruption of the mitochondrial respiratory chain, the cellular oxygen consumption of treated C2C12 cells was assessed with a SeahorseXF24 analyzer. Basal, leak, and maximal respiration of the cells is shown in Figure 3. After treatment for 24 h, a decrease in basal respiration was observed for the pyrovalerone derivatives naphyrone and α-PVP, starting at 500 μM. In addition, both drugs decreased the maximal respiratory capacity of the cells at equal concentrations. α-PVP seemed to decrease the maximal oxygen consumption already at 200 μM, statistical significance was, however, not reached. The remaining drugs did not impair cellular respiration in the investigated concentration range with the exception of MDMA, which decreased the maximal respiratory capacity at concentrations ≥1 mM.

Almost identical results were obtained after incubation of C2C12 myoblasts for 1 h (Appendix A). Similar to the incubations for 24 h, methylone, mephedrone, 3-MMC and MDPV did not impair cellular oxygen consumption. In contrast to the 24 h study, MDMA did not inhibit cellular oxygen consumption, whereas α-PVP and naphyrone inhibited oxygen consumption after 1 h starting at 1000 and 200 µM, respectively.

### 2.3. Activity of Enzyme Complexes of the Mitochondrial Electron Transport Chain

MDMA, and the pyrovalerone cathinones α-PVP and naphyrone, disrupted cellular respiration in C2C12 myoblasts in addition to decreasing ATP levels and cell viability. Therefore, the activity of the enzyme complexes of the mitochondrial electron transport chain was investigated in more detail using a high-resolution respirometry system (Oxygraph-2k). For each drug, the lowest concentration at which an impairment of cellular oxygen consumption became apparent was chosen for further investigation of the activity of the enzyme complexes. As shown in Figure 4, MDMA inhibited complex I (glutamate) and III (duroquinol), α-PVP complex I (glutamate) and naphyrone complex II (succinate). None of the compounds investigated impaired the activity of complex IV (TMPD/ascorbate). In the presence oligomycin, no stimulation of the respiration was observed compared to control values, excluding the uncoupling of oxidative phosphorylation. Interestingly, the oxygen consumption in the presence of succinate was low compared to the other substrates, indicating a low activity of complex II in C2C12 myoblasts.

### 2.4. Mitochondrial Superoxide Production

The accumulation of reactive oxygen species (ROS) was one of the consequences of the impairment of the mitochondrial respiratory chain. All the drugs caused an increase in superoxide in a concentration dependent manner (Figure 5). Consistent with ATP depletion, naphyrone caused elevated superoxide concentrations at the lowest concentration (200 μM) among all drugs investigated. 3-Methylmethcathinone induced significant increase in superoxide production at 500 μM, and the remaining drugs at 1000 μM.

It was surprising to observe that all the investigated compounds caused an increase in the mitochondrial superoxide concentration. One reason could be a low mitochondrial antioxidative capacity in mitochondria from C2C12 myoblasts. We therefore determined the protein content of superoxide dismutase 2 (SOD2), the enzyme which degrades intramitochondrial superoxide [40], in C2C12 myoblasts and myotubes after differentiation. As shown in Appendix A, the protein content of SOD2 was approximately 3 times lower in myoblasts than in myotubes, suggesting that impaired degradation could play a role in the accumulation of mitochondrial superoxide as observed in the presence of the compounds investigated.

## 3. Discussion

All of the investigated synthetic cathinones, as well as MDMA, depleted the cellular ATP pool and impaired cell membrane integrity in C2C12 myoblasts, but at concentrations (≥50 μM) that are not reached with ingestion of the typical doses. However, patients with preexisting mitochondrial diseases, or who ingested other myotoxic drugs, may be more sensitive to these drugs. Patients who ingested suprapharmacological doses may reach such concentrations [41]. Nevertheless, such studies are helpful to investigate the mechanisms of toxicity, for instance, mitochondrial toxicity [42]. Similar to previous results in HepG2 cells [30], all synthetic cathinones impaired cell membrane integrity at almost identical concentrations. Naphyrone was the most toxic compound investigated, which may be related to the reactivity of the naphthyl group. The direct comparison between MDMA and methylone (which differ only in the β-keto group in the side chain, see Figure 1) suggests no major role of the β-keto group regarding the toxicity in C2C12 myoblasts.

A decrease in ATP that precedes cell membrane integrity loss after drug treatment is reported to indicate that the mitochondria, as primary ATP-producing organelles, are targeted by these drugs [38,43]. Considering the IC50 values for ATP depletion and membrane integrity, a trend for ATP depletion preceding loss of cell membrane integrity was visible for 3-MMC, mephedrone, α-PVP, MDPV and naphyrone, but not for MDMA and methylone (Table 1). This possible indicator of mitochondrial toxicity was confirmed by the more detailed mitochondria-specific investigations, and for this reason, cellular respiration was assessed after exposing C2C12 myoblasts to the test drugs. Xenobiotics may disrupt cellular respiration either by inhibition of enzyme complexes of the electron transport chain, or by uncoupling the activity of the electron transport chain from ATP synthesis, or a combination thereof [38,43,44,45]. Such events may then lead to a decrease in ATP production, increased production of superoxide, disruption of Δψm, and eventually cell death by apoptosis or necrosis. The investigations of cellular respiration after pretreatment with the synthetic cathinones for 1 or 24 h revealed that only pretreatment with the two pyrovalerone derivatives α-PVP and naphyrone, but not with the pyrovalerone derivative MDPV or any of the other investigated substances, decreased basal respiration. In addition to the decrease in basal respiration, α-PVP and naphyrone impaired the maximal respiratory capacity at similar concentrations (500 to 1000 μM). Interestingly, MDMA was only toxic after 24 h but not at 1 h of incubation, suggesting that the toxicological mechanisms may be different between cathinones and MDMA.

Compared to vehicle control treatment, no difference in proton leak was observed for any of the tested compounds in the investigated concentration range, suggesting that the compounds did not directly uncouple oxidative phosphorylation. However, uncoupling of oxidative phosphorylation has previously been identified as an important contributor to MDMA-induced hyperthermia, which is attributable to activation (by phosphorylation) of the uncoupling protein 3 (UCP-3) in skeletal muscle [35,37,46]. The activation of UCP-3 has been proposed to be mediated by norepinephrine release, following activation of the sympathetic nervous system by MDMA [47]. Since we carried out our investigations in vitro, this mechanism was impossible, possibly explaining the lack of uncoupling of oxidative phosphorylation by the compounds investigated. Many cathinone designer drugs exert similar effects on the noradrenergic system as MDMA [10,11,14], and may therefore potentially cause hyperthermia mediated through UCP-3 activation in vivo as well. However, a direct uncoupling effect of the prescription cathinone bupropion has previously been observed in human hepatoma cells [30], suggesting that some synthetic cathinones that were not investigated in the current study can have a direct uncoupling effect in skeletal muscle cells. Furthermore, since the mechanisms of cathinone-induced hyperthermia may differ from those of MDMA [48], they should be investigated in more detail in future studies.

The effect of MDMA, α-PVP, and naphyrone on the enzyme complexes of the mitochondrial electron transport chain was investigated in more detail to assess whether impairment of these complexes may be the underlying mechanism leading to disturbance of the electron transport chain. As shown in Figure 4, all three compounds impaired the activity of enzyme complexes of the electron transport chain. This observation may explain the decrease in the cellular oxygen consumption and ATP production observed for these substances.

The mitochondrial respiratory chain is a major source of ROS and complex I and III, and has been shown to be potential sites for ROS generation [49,50,51]. We therefore expected to detect an increase in mitochondrial ROS production for MDMA, α-PVP and naphyrone. To our surprise, we not only observed an increase in the cellular ROS content for these three compounds, but also for the other compounds investigated, which were not found to impair the electron transport chain. A possible reason could be a low antioxidative capacity in C2C12 myoblasts as suggested by a low expression of SOD2, the enzyme responsible for mitochondrial breakdown of superoxide [40].

The question arises as to why we observed ATP depletion for 3-MMC, mephedrone, MDPV and methylone without detecting mitochondrial dysfunction. In the presence of glucose, ATP can also be derived from glycolysis, which can also be impaired by toxicants [39]. Furthermore, the IC50 for impairment of membrane integrity and ATP depletion were close, suggesting that ATP depletion could have been a secondary event after disruption of membrane integrity.

## 4. Materials and Methods

### 4.1. Chemicals

3-MMC, MDMA, MDPV, methylone, mephedrone, and α-PVP were purchased as hydrochlorides from Lipomed (Arlesheim, Switzerland) with high-performance liquid chromatography (HPLC) purity >98.5%. Naphyrone hydrochloride was available from a previous study [11], after originally being synthesized as described by Meltzer et al. [52]. Drug stocks were made in autoclaved deionized water and were freshly prepared for each assay.

### 4.2. Cell Culture

C2C12 myoblasts were obtained from the American Type Culture Collection and kindly provided by Novartis (Basel, Switzerland). The cells were cultured in Dulbecco’s Modified Eagle’s Medium (DMEM) supplemented with 10% heat-inactivated fetal bovine serum at 37 °C in a humidified 5% CO_2_ atmosphere [53]. When 70–80% confluency was reached, the cells were passaged with TrypLE Express reagent (Invitrogen, Basel, Switzerland). Undifferentiated C2C12 myoblasts were chosen for the current study due to the higher sensitivity to xenobiotics compared to differentiated C2C12 myotubes.

### 4.3. Plasma Membrane Integrity

As marker of membrane toxicity, cell membrane integrity loss expressed as adenylate kinase (AK) release, was determined with the ToxiLight BioAssay kit (Lonza, Basel, Switzerland) [38]. In brief, C2C12 myoblasts were seeded at a density of 50,000 cells per well in a 96-well plate. After attachment, the cells were treated with 100 μL of culture medium containing the test drugs (naphyrone at the concentrations of 10–500 μM, the remaining drugs at concentrations of 50–2000 μM). Triton X-100 (0.1%) was included as positive control inducing cell lysis. After 24 h, 20 μL of cell supernatant was transferred to a luminescence compatible 96-well plate and 100 μL of AK detection reagent was added to each well. After shaking of the plate at room temperature for 5 min, luminescence was measured with a Tecan M200 Pro Infinity plate reader (Tecan, Männedorf, Switzerland). The percentage of intact cells (no cell membrane integrity loss) was calculated in relation to untreated cells and cells treated with Triton X-100, which were determined to represent 100% and 0% intact cells, respectively.

### 4.4. ATP Content

The intracellular ATP content representing the metabolic cell activity was measured using a CellTiter-Glo kit (Promega, Dübendorf, Switzerland) [38]. C2C12 myoblasts were seeded at a density of 50,000 cells per well in a 96-well plate. After attachment, the cells were treated with 100 μL of culture medium containing the test drugs (naphyrone at the concentrations of 10–500 μM, the remaining drugs at concentrations of 50–2000 μM). Triton X-100 (0.1%) was included as positive control inducing cell lysis. After 24 h, 50 μL of supernatant was removed from each well before 50 μL of CellTiter-Glo Reagent was added to the remaining cell culture medium at a ratio of 1:1 (*v*/*v*). After shaking of the plate at room temperature for 15 min, luminescence was measured with a Tecan M200 Pro Infinity plate reader (Tecan, Männedorf, Switzerland).

### 4.5. Oxygen Consumption

To determine whether synthetic cathinones target mitochondria, the oxygen consumption rate (OCR) of C2C12 myoblasts was measured with a Seahorse XF24 analyzer (Seahorse Biosciences, North Billerica, MA, USA). Briefly, 50,000 C2C12 cells per well were seeded in a Seahorse XF 24-well cell culture microplate coated with poly-d-lysine. The attached cells were then treated with a culture medium containing the test drugs (naphyrone at the concentrations of 50–500 μM, the remaining drugs at concentrations of 200–2000 μM) for 24 h. Thereafter, the culture medium was removed and the cells were washed twice with unbuffered DMEM (4 mM l-glutamate, 1 mM pyruvate, 4.1 g/L glucose, 63.3 mM sodium chloride, pH 7.4). After the second washing step, 750 μL of unbuffered DMEM was added to each well and the cells were incubated in a CO_2_-free incubator for 40 min at 37 °C before they were transferred into the XF24 analyzer. After measuring the basal oxygen consumption rate, 1 μM oligomycin (inhibitor of F0F1ATPase), 1 μM carbonyl cyanide-4-(trifluoromethoxy)phenylhydrazone (FCCP; uncoupling agent), and 1 μM rotenone (complex I inhibitor) were successively added to the cells in order to measure proton leak, maximal cellular respiration, and extramitochondrial respiration, respectively. To adjust the OCR to the protein content, the cells were first fixed with 100 μL 50% (*w*/*v*) trichloroacetic acid added directly to the assay medium of each well for 1 h at 4 °C. Thereafter, the cells were washed with deionized water and treated with 0.4% (*w*/*v*, in 1% [*v*/*v*] acetic acid) sulforhodamine B. After 20 min, the cells were rapidly washed with 1% (*v*/*v*) acetic acid and the incorporated dye was solubilized with 100 μL of 10 mM TRIS base. The absorbance was then measured at 490 nm to obtain the amount of protein in relation to vehicle control.

### 4.6. Activity of Specific Enzyme Complexes of the Mitochondrial Electron Transport Chain

The activity of complex I and complex III of the mitochondrial respiratory chain was analyzed with an Oxygraph-2k high-resolution respirometer equipped with DataLab software (Oroboros Instruments, Innsbruck, Austria). Briefly, C2C12 cells were treated with either MDMA (1 mM), α-PVP (200 μM), or naphyrone (500 μM). After 24 h exposure, cells were resuspended in mitochondrial respiration medium MiR05 (0.5 mM EGTA, 3 mM magnesium chloride, 20 mM taurine, 10 mM potassium dihydrogen phosphate, 20 mM HEPES, 110 mM sucrose, 1 g/L fatty-acid free bovine serum albumin, and 60 mM lactobionic acid, pH 7.1) and a volume of 200 μL containing 2 million cells was transferred into the pre-calibrated Oxygraph chamber [54]. To permeabilize the cells, digitonin (10 μg per million cells) was added to the chamber. Then, the activity of complex I (NADH-CoQ reductase) was measured by adding L-glutamate and malate (10 and 2 mM, respectively) as substrates of the electron transport chain, followed by the addition of adenosine-diphosphate (ADP; 2.5 mM). To determine the activity complex III (CoQH2-cytochrome c reductase), complex I was inhibited with rotenone (0.5 μM) and duroquinol (0.5 mM) and was then added as a substrate of complex III.

### 4.7. Mitochondrial Superoxide Production

The MitoSOX red superoxide (O^2−^) indicator (Invitrogen, Basel, Switzerland) was used for the assessment of oxidative stress in C2C12 cells treated with the test drugs. MitoSOX red is a live-cell permeant fluorogenic dye that targets the mitochondria and exhibits red fluorescence upon oxidation by superoxide. In brief, 50,000 cells were seeded at a density of 50,000 cells per well in a black clear-bottom 96-well plate. After attachment, the cells were treated with 100 μL of culture medium containing the test drugs (naphyrone at the concentrations of 10–500 μM, the remaining drugs at concentrations of 50–2000 μM). Amiodarone (20 μM) was included as a positive control. After 24 h, the cells were treated with 2.5 μM of MitoSOX reagent (100 μL per well) and incubated for 20 min at 37 °C and protected from light. Fluorescence was measured at 510/580 nm using a Tecan M200 Infinite Pro Infinity plate reader (Männedorf, Switzerland). The protein content was assessed with the Pierce Protein Assay kit (Thermo Scientific, Wohlen, Switzerland) and the fluorescence signal adjusted to protein content.

### 4.8. Statistics

Data are presented as mean ± SEM of at least three independent experiments and GraphPad Prism 7 (GraphPad Software, La Jolla, CA, USA) was used for statistical analysis. The activity of the enzyme complexes of the mitochondrial electron transport chain was compared to vehicle control with an unpaired two-tailed Student’s *t*-test. For the remaining assays, statistical differences between control and test drugs were calculated with ANOVA followed by Dunett’s test. *p* values below 0.05 were considered to be statistically significant.

## 5. Conclusions

All investigated synthetic cathinones, as well as MDMA, depleted ATP and elicited cell membrane integrity loss, and increased superoxide levels in C2C12 myoblasts in a concentration-dependent manner. The two pyrovalerone derivatives, α-PVP and naphyrone, additionally impaired basal and maximal cellular respiration, which suggests mitochondrial dysfunction. Therefore, in addition to sympathetic nervous system effects and strenuous muscle exercise, direct effects of some cathinones on skeletal muscle mitochondria may contribute to myotoxicity in susceptible cathinone users.

## Figures and Tables

**Figure 1 ijms-20-01561-f001:**
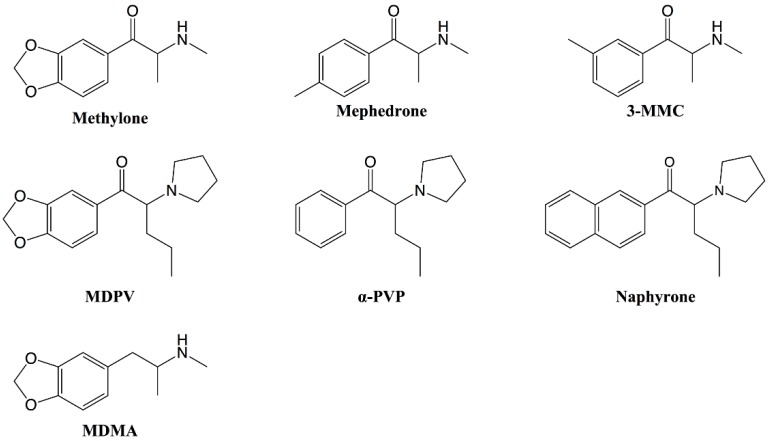
Structures of the synthetic cathinones included in the study and the reference substance 3,4-methylenedioxymethamphetamine (MDMA). MDPV: 3,4-methylenedioxypyrovalerone; α-PVP: alpha-pyrrolidinovalerophenone; 3-MMC: 3-methylmethcathinone

**Figure 2 ijms-20-01561-f002:**
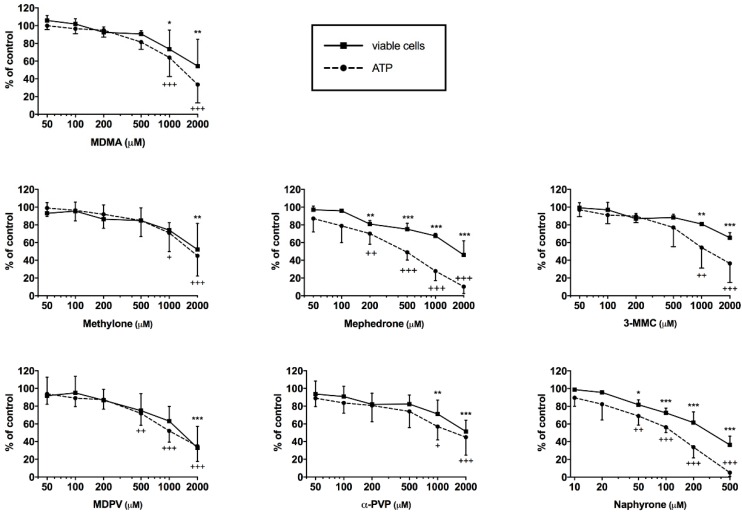
Intracellular ATP content and viable cells expressed by maintained cell membrane integrity after drug exposure for 24 h in C2C12 cells. Data are expressed as mean ± SEM of at least three independent experiments. Drug treatments were compared to vehicle control with ANOVA followed by Dunnett’s test. Significance levels for cell viability are given as * *p* < 0.05, ** *p* < 0.01, *** *p* < 0.001. Significance levels for ATP content are given as + *p* < 0.05, ++ *p* < 0.01, +++ *p* < 0.001.

**Figure 3 ijms-20-01561-f003:**
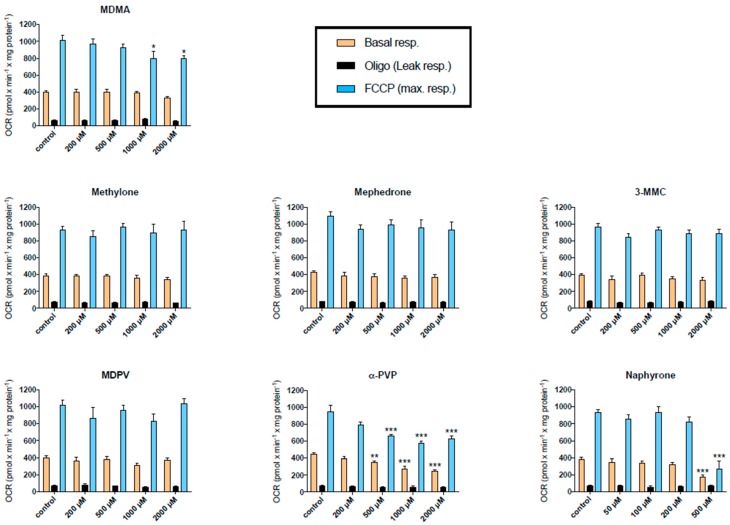
Oxygen consumption rate (OCR) in C2C12 cells after 24 h drug exposure. Basal respiration, leak respiration, and maximal respiration are expressed as mean ± SEM of at least three independent experiments. Drug treatments were compared to vehicle control with ANOVA followed by Dunnett’s test. Significance levels are given as * *p* < 0.05, ** *p* < 0.01, *** *p* < 0.001.

**Figure 4 ijms-20-01561-f004:**
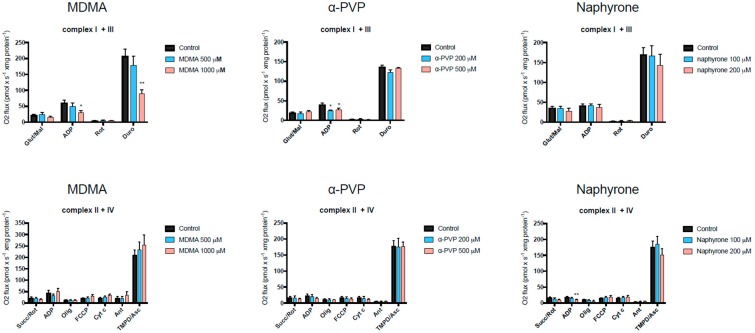
Effect on the activity of the enzyme complexes of the mitochondrial electron transport chain in C2C12 cells measured using a Oxygraph-2k-high-resolution respirometer. Data are expressed as mean ± SEM of at least four independent experiments. Treatment was compared to vehicle control with an unpaired two-tailed Student’s *t*-test. Significance levels are given as * *p* < 0.05, ** *p* < 0.01

**Figure 5 ijms-20-01561-f005:**
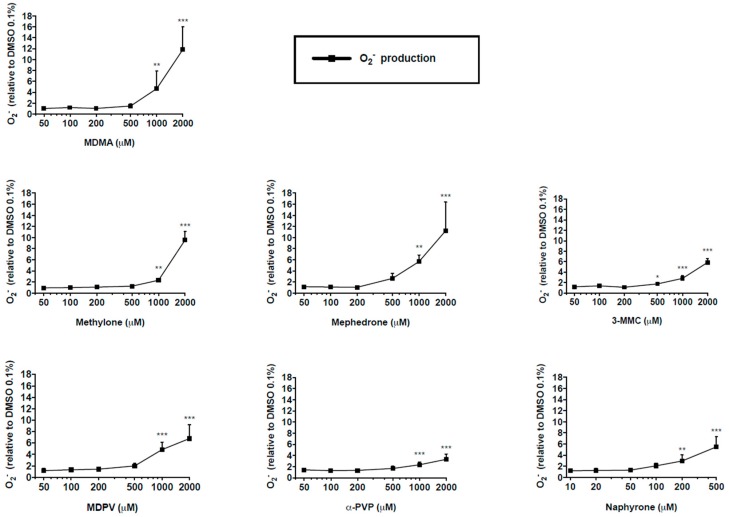
Mitochondrial superoxide production in C2C12 cells after 24 h drug treatment. Data are expressed as mean ± SEM of at least three independent experiments compared to vehicle control and were analyzed with ANOVA followed by Dunett’s test. Significance levels are given as * *p* < 0.05, ** *p* < 0.01, *** *p* < 0.001.

**Table 1 ijms-20-01561-t001:** Quantification (IC_50_) of membrane toxicity and ATP depletion by the investigated synthetic cathinones and MDMA. Unit for IC_50_ is mM.

	MDMA	3-MMC	Mephedrone	Methylone	α-PVP	MDPV	Naphyrone
Membrane toxicity	0.74	>2	>2	1.10	1.04	>2	0.047
ATP depletion	0.88	1.08	0.53	1.16	0.94	1.01	0.021

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
