# Peer review of "Molecular Toxicological Mechanisms of Synthetic Cathinones on C2C12 Myoblasts"

_ijms, 2019, doi:10.3390/ijms20071561_

Round 1

Reviewer 1 Report

Line 33: Catha edulis shrub should be in italic

Line 35: the sentence is confusing: "These substances are ..."

Line 43: it would be appropiate to include in the introduction a brief description of the sympathomimetic toxicity 

Line 95: Figure 3: the bars should be in colors

Line 118: Figure 4: the same of figure 3.  

Line 130: Figure 5: improve the legend.  It should explain briefly in the legend how the percentages of the y-axis were calculated

Line 218-219: reference?

Line 227: reference?

Author Response

Line 33: Catha edulis shrub should be in italic: we have done that. 

Line 35: the sentence is confusing: "These substances are ...": we have modified this passage. 

Line 43: it would be appropriate to include in the introduction a brief description of the sympathomimetic toxicity: thank you for this remark; we have included the main symptoms of sympathomimetic toxicity. 

Line 95: Figure 3: the bars should be in colors: we have done that. 

Line 118: Figure 4: the same of figure 3: we have done that. 

Line 130: Figure 5: improve the legend. It should explain briefly in the legend how the percentages of the y-axis were calculated: the axis label of the ordinate was wrong, thank you for communicating this mistake. It must read “relative to control” instead of % of positive control. We have corrected the ordinate accordingly. 

Line 218-219: reference: we have included a reference for the culture of myoblasts (new ref. 52).

Line 227: reference: we performed the assay according to ref. 38, which we have included.

Reviewer 2 Report

ijms-447678

Comments to authors:

This manuscript entitled “Molecular toxicological mechanisms of synthetic cathinones on C2C12 myoblasts” provided the safety information of synthetic cathinones. This manuscript is well structured and contains useful information for the T. camphoratus extract. There are some questions below, please consider the comments.

Minor comments:

1.         In clinic, how high concentration in plasma of synthetic cathinones will cause myotoxicity in human?

2.         Data showed synthetic cathinones could induce cytotoxicity in C2C12 myoblasts, but at concentrations (≥50 μM) that are not reached with the typical doses ingested. How to transfer the intoxic concentrations in vitro to human?

3.         The biomarkers of the myotoxicity may include as CK and LDH activities. Does this study also examine these enzymatic activities or morphological changes in C2C12 myoblasts?

4.         Please check the name of refs, such as: …J Med Toxicol or Archives of Toxicology, abbreviation or not?

Author Response

1. In clinic, how high concentration in plasma of synthetic cathinones will cause myotoxicity in human? To the best of our knowledge, there are currently no reported plasma levels in patients with rhabdomyolysis caused by the use of synthetic cathinones. However, reported plasma levels of rhabdomyolysis cases linked to the use of MDMA were >8 mg/l (»40 µM) (Mayo Clin Proc 2001;76:112-3, ref. [33]) and 1.5 mg/l MDMA (8 µM) (J Emerg Med 2012;42:655-8, ref. [36]). Since the myotoxic potential of the synthetic cathinones investigated in the study was similar to MDMA, we may assume that rhabdomyolysis could occur at similar concentrations. In post-mortem cases, synthetic cathinone concentrations up to 22 mg/l (»100 µM) have been reported (Forensic Sci Int 2010;202:e62-63).

2. Data showed synthetic cathinones could induce cytotoxicity in C2C12 myoblasts, but at concentrations (≥50 μM) that are not reached with the typical doses ingested. How to transfer the intoxic concentrations in vitro to human? As you can see in the answer to question 1, concentrations up to 100 µM have been reported in intoxicated patients. We agree, however, that patients with rhabdomyolysis have typically lower concentrations than 100 µM. It is possible that the effect on body temperature occurs at lower concentrations, that other drugs contribute or that the patients have susceptibility factors rendering them more sensitive as for instance impaired mitochondrial function. We discuss that on line 147-150 of the mansucript. 

3. The biomarkers of the myotoxicity may include as CK and LDH activities. Does this study also examine these enzymatic activities or morphological changes in C2C12 myoblasts? We measure AK release and ATP as markers of cytotoxicity. AK release is similar to the release of LDH and CK and reflects the intactness of the plasma membrane. In our view, there should not be a difference between AK release and the release of LDH and CK; the mechanism leading to the release from cells is the same (intactness of plasma membrane). In addition, we determined the cellular ATP content, which reflects the energy metabolism in C2C12 myoblasts, but also the intactness of the plasma membrane. The situation where the plasma membrane is intact but the cellular ATP content is decreased reflects impaired ATP production, which is in most cases mitochondrial. Our determinations therefore provide more information than “only” CK and LDH. 

4. Please check the name of refs, such as…J Med Toxicol or Archives of Toxicology, abbreviation or not? Thank you for this comment. We have corrected that.
